# A graphical exploration of the relationship between parasite aggregation indices

R. McVinish[1], R. J. G. Lester [2]*

**1** School of Mathematics and Physics, The University of Queensland, Brisbane, Australia, **2** School of the Environment, The University of Queensland, Brisbane, Australia

* r.lester@uq.edu.au

**Data Availability Statement:** "All relevant data are within the paper and its Supporting Information files."

**Funding:** The author(s) received no specific funding for this work.

## Abstract

The level of aggregation in parasite populations is frequently incorporated into ecological studies. It is measured in various ways including variance-to-mean ratio, mean crowding, the $k$ parameter of the negative binomial distribution and indices based on the Lorenz curve such as the Gini index (Poulin's D) and the Hoover index. Assuming the frequency distributions follow a negative binomial, we use contour plots to clarify the relationships between aggregation indices, mean abundance and prevalence. The contour plots highlight the non-linear nature of the relationships between these measures and suggest that correlations are not a suitable summary of these relationships.

## 1. Introduction

Living organisms are typically not evenly distributed across their range; individuals are clustered together, a phenomenon termed aggregation. In free-living animals this may have advantages for reproduction, foraging success and/or security against predators. In parasites, the benefits of aggregation are less clear but include opportunities for cross fertilisation, though could also influence parasite survival and determine whether a host population becomes extinct [1].

The origins of parasite aggregation were summarised by Poulin as heterogeneity in acquisition and heterogeneity in susceptibility [2]. Heterogeneity in acquisition may be amplified by the complex life cycles of some parasites [3, 4]. Heterogeneity in susceptibility may be amplified by artificial methods used to control parasites. Such control can lead to a higher concentration of parasites in fewer individuals which may require modification of the control strategy [5–7].

But how to measure aggregation? Though it is a widespread and significant phenomenon, its concept in parasitology is poorly defined [8, 9]. Many authors have sought to compare the levels of aggregation between different parasite populations, sometimes developing elaborate formulae [10–12]. One frequently used index is the variance to mean ratio (VMR). Another common index dates from Crofton [13] who pointed out that parasites are frequently distributed according to a negative binomial distribution; the 'k' of the negative binomial is now sometimes referred to as the 'aggregation parameter'. Closely related to VMR and $k$ are mean crowding and patchiness [14] which can be interpreted as a measure of the competitive experience of parasites within a host [15].

**Competing interests:** The authors have declared that no competing interests exist.

Two indices that have been used recently are derived from the Lorenz curve [16]. The Gini index [17], a quantification of inequality in economics, was proposed for use in parasitology by Poulin [18] as 'Discrepancy D', sometimes referred to as 'Poulin's D'. It has been widely applied in parasitology [19–21]. Another index derived from the Lorenz curve, the Hoover index (aka Pietra index), was used recently in [9, 22].

Here we present 'contour plots' that show the relationship between these two aggregation indices, and mean abundance and prevalence. The indices were calculated directly from the parameters of the negative binomial distributions. The plots demonstrate the relationships are non-linear and not readily interpreted by correlation analysis (cf. [23]).

## 2. Contour plots

Contour plots show combinations of two indices, specified on the vertical and horizontal axes, that give rise to similar values of the third index. The technique, developed in the 16$^{th}$ century [24], is widely used in other disciplines but rarely in parasitology.

Our analysis assumed that parasite burden is adequately modelled by a negative binomial distribution [2, 13, 23, 25, 26]. Following the typical practice in parasitology, we parameterised the negative binomial distribution in terms of mean abundance, $m$, and the parameter $k$ which controls the shape of the distribution. We did not make any assumption on the distribution of $m$ and $k$. We used the range of values for $m$ and $k$ suggested by the extensive data of Shaw and Dobson [25]. Their values for $m$, $k$ and prevalence are superimposed on several of the contour plots as dot points.

To construct a contour plot of an aggregation index against $m$ and $k$, we expressed the aggregation index as a function of $m$ and $k$. The population values of several indices can be expressed simply in terms of $m$ and $k$: prevalence = $1 - (k/(k + m))^k$, VMR = $1 + m/k$, mean crowding = $m + m/k$, and patchiness = $1 + 1/k$. The Gini and Hoover indices lack simple expressions in terms of $m$ and $k$ however, they can still be evaluated numerically. The Hoover index can be expressed in terms of $m$ and $k$ by applying Lemma 5.3.3 of Arnold & Sarabia [27],

$$H = F(m; k, m) - F\left(m - 1; k + 1, m + \frac{m}{k}\right),$$

where $F(x; k, m)$ is the cumulative distribution function of the negative binomial distribution with $k$ and mean $m$ evaluated at $x$. Further details are given in the S1 Supplementary Material in S1 File. The cumulative distribution function of the negative binomial distribution, $F$, is available in statistical packages such as R [28]. The Gini index can be expressed as

$$G = \left(1 + \frac{m}{k}\right) {}_2F_1\left(k + 1, \frac{1}{2}, 2; -4\frac{m}{k}\left(1 + \frac{m}{k}\right)\right),$$

where ${}_2F_1$ is the Gaussian hypergeometric function [29]. This can be evaluated in R using the hypergeo package [30]. Taking the values for $m$ and $k$ reported in Shaw & Dobson [25] as a guide, we specified the range of values for $m$ and $k$, with $m$ ranging between 0.1 and 5200 and $k$ ranging between 0.001 and 16.5. The index is then evaluated on a grid of values in the $(k, m)$ plane. The contour plots were produced in R using the ggplot2 package [31]. Since the values for $m$ and $k$ reported in Shaw & Dobson [25] are heavily right skewed and the indices undergo the most rapid change for values of $m$ and $k$ around 0.1–1.0, log scaling has been applied to these variables to improve clarity of the plot.

We also employed contour plots to examine the relationship between the indices, $m$ and prevalence. This required first determining values of $m$ and prevalence that are consistent with the negative binomial distribution. A pair of values of $m$ and prevalence are consistent with

the negative binomial distribution if the equation

$$\text{prevalence} = 1 - (k/(k + m))^k$$

has a solution for $k > 0$. This equation has no solution if $m + \ln(1\text{-prevalence}) < 0$. On the other hand, if $m + \ln(1\text{-prevalence}) > 0$, then the equation has a unique solution, and the solution was found numerically using the uniroot function in R. For all pairs in the grid on the (prevalence, $m$) plane that are consistent with the negative binomial distribution the aggregation indices were evaluated using the expressions in terms of $m$ and $k$. The contour plots were again produced in R using the ggplot2 package [31]. Regions of $m$ and prevalence that were inconsistent with a negative binomial distribution are represented as white in the contour plot.

## 3. Relationship between mean abundance, *k* and prevalence

The relationship between $m$, $k$ and prevalence in wild populations has been examined by several authors with conflicting results [18, 25, 32–34]. While the expression of prevalence in terms of $m$ and $k$ is sufficiently simple to analyse, it is still instructive to construct the contour plot (Fig 1, left). In it, each colour represents a region of values of $m$ and $k$ that give rise to similar values of prevalence.

As prevalence is increasing in both $m$ and $k$, this leads to contours that are roughly L-shaped on the range of $m$ and $k$ plotted. We see that prevalence is small (dark band) when either $m$ or $k$ are small, and that prevalence is large (yellow band) when both $m$ and $k$ are large. The contours also show that there is a non-linear relationship between $m$ and $k$ when prevalence is considered fixed. Contours become almost parallel to the horizontal axis as $k$ increases, a consequence of $\lim_{k\to\infty}$ prevalence $= 1 - e^{-m}$. On the other hand, the contours continue to

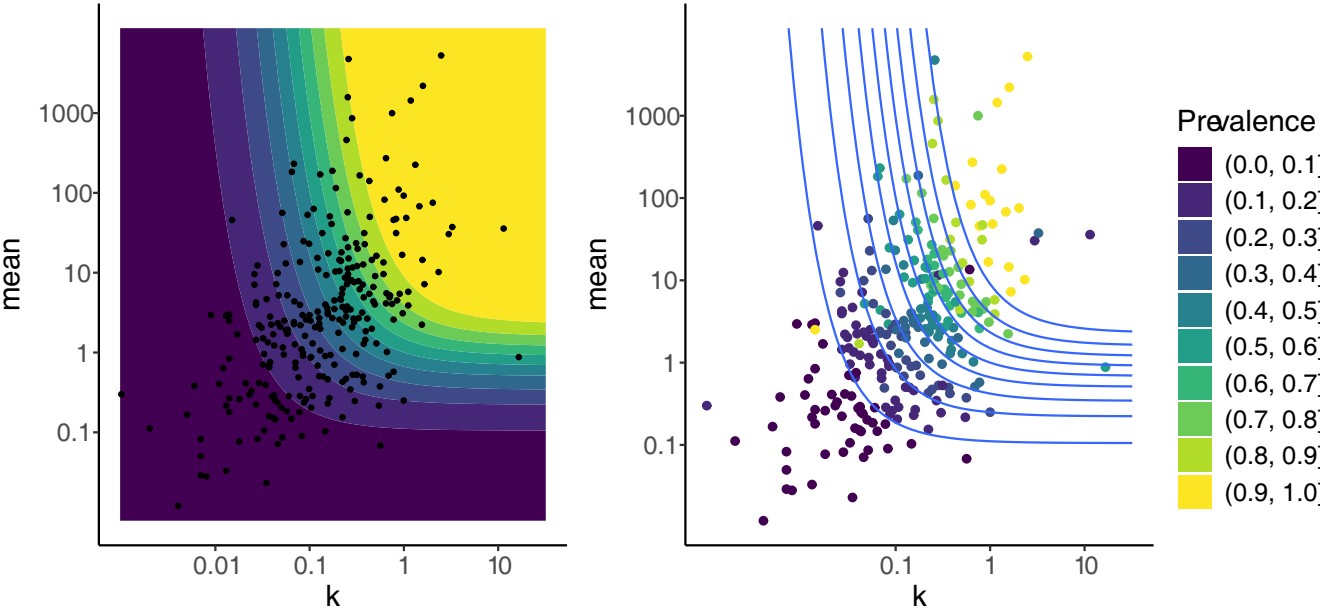

**Fig 1.** (Left) Contour plot showing prevalence levels from zero to 1 for values of $m$ and $k$. Each colour band represents a region of values of $m$ and $k$ that give rise to similar values of prevalence. The axes for both the mean and $k$ are on the log scale. Dot points are actual data from Shaw and Dobson [25]. (Right) Scatter plot of mean against $k$ with prevalence data from Shaw & Dobson. Lighter colours indicate higher prevalence level. The lines are from the contour plot (left). In general, there is good agreement between the observed prevalences and the contours. Exceptions may be from samples that did not conform to a negative binomial.

move left as $m$ increases, a consequence of $\lim_{m\to\infty}$ prevalence $= 1$. The contour plot shows that the rate at which prevalence approaches one as $m$ increases is slow when $k$ is small.

If we restrict our attention to a single-coloured band, i.e. those values of $m$ and $k$ giving rise to similar values of prevalence, we see that, after controlling for prevalence, there is a negative relationship between $m$ and $k$. This relationship is forced by the negative binomial distribution, so it will hold true in natural systems to the extent that those systems are well modelled by the negative binomial distribution. The different widths of the contour lines show the non-linearity of the relationship between $k$, $m$ and prevalence.

The dot points represent estimates of $m$ and $k$ from the 269 parasite-host systems reported in [25]. Although several parasite-host systems lie in a region of very high prevalence (both $m$ and $k$ large) or very small prevalence (either $m$ or $k$ small), many others occupy a region of the parameter space where a moderate change in the parameter values would result in a significant change in prevalence assuming a negative binomial distribution.

As Shaw & Dobson reported prevalence values in their review [25], it is possible to compare these with the prevalence values implied by the negative binomial distribution (Fig 1, right). In general, there is good agreement; most points within a given contour having the same colour. This demonstrates the accuracy of the contour plots to interpret relationships in real life situations. The few points where the observed prevalences don't agree with that determined by the negative binomial could be because these distributions did not conform to a negative binomial.

## 4. Relationship of Hoover and Gini indices with mean abundance, *k* and prevalence

Contour plots of the Hoover index and Gini index as functions of $m$ and $k$ are shown in Fig 2 left and right. The contour plots are qualitatively very similar and share some similarities with the contour plot of prevalence (Fig 1). Both Hoover and Gini indices decrease in both $m$ and $k$,

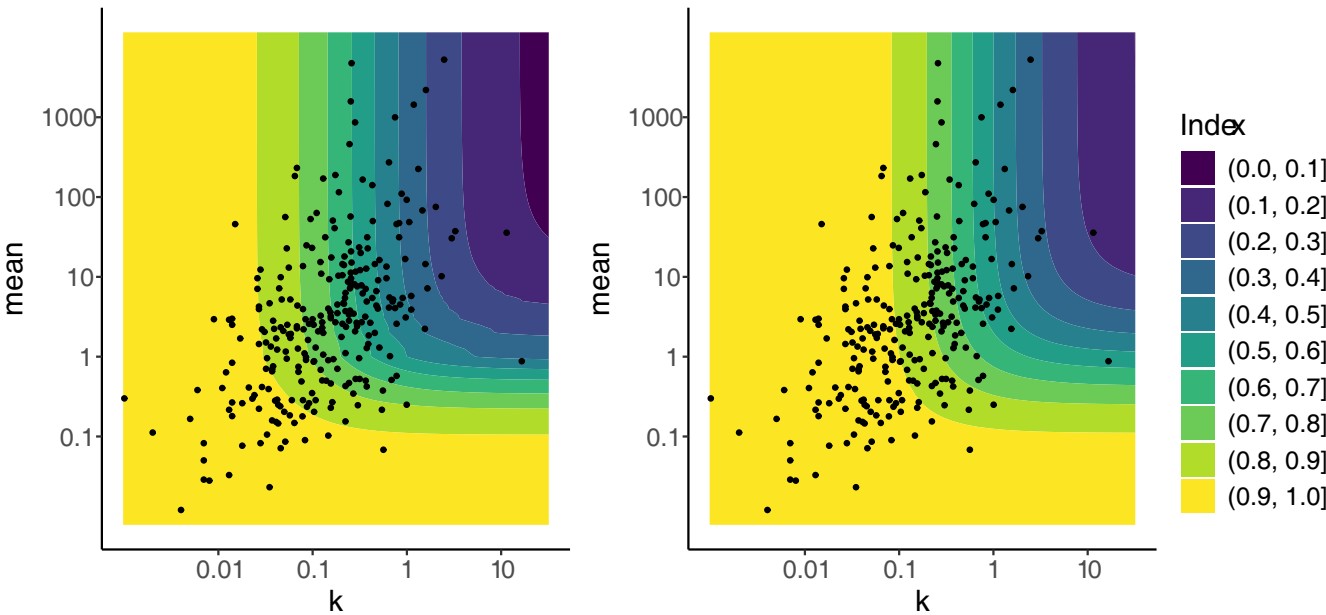

**Fig 2.** Contour plots of Hoover index (left) and Gini index (right) as functions of $m$ and $k$. Contour lines are shown from 0 to 1, i.e. from least aggregated (darkest band) to most aggregated (yellow band). Comparing the graphs, the Gini index is always larger than the Hoover index and has a smaller range over the region of values for $m$ and $k$ observed in wild populations (dot points).

taking values close to one when either *m* or *k* were small, and taking values close to zero when both *m* and *k* were large. The contours are L-shaped becoming almost parallel to the horizontal axis as *k* increases and almost parallel to the vertical axis as *m* increases.

The plots show both indices display some stability with respect to the mean as the bands are parallel to the vertical axis over a wide range of *m* and *k*. Restricting our attention to the Hoover index (Fig 2, left), we see that for *m* > 5 the value of the index is largely determined by the size of *k*. For *m* < 5 the value is less affected by *k* but more affected by *m*, as indicated by the number of contours crossed as *m* decreases. For example, starting from *k* = 1 and *m* = 6, as *m* decreases the value of the index increases quickly crossing several contours from 0.4 to 1. On the other hand, when *m* increases from the same point (1,6) the index stays in the same colour band and there is little change in the Hoover value (0.4 to 0.5). For many of the parasite-host systems reported in [25], shown on the figure as dot points, an increase in, that is moving the points vertically on the contour plot, does not appear to impact the Hoover index since the point would remain in the same-coloured region. On the other hand, in many of the samples, a moderate change in *k*, that is moving the point horizontally, has a large impact on the Hoover index. Similar behaviour is observed in the contour plot of the Gini index (Fig 2, right), with the Gini index appearing to be even less affected by changes in *m*. Further investigation of the stability of the plots is given in S3 Supplementary Material in S1 File.

There are two noticeable differences between the contour plots for the Hoover and Gini indices (Fig 2). Firstly, the Gini index is always larger than the Hoover index [27, 35]. This causes the Gini index to have a smaller range over the region of values for *m* and *k* observed in wild populations. Second, the contours of the Hoover index are not smooth, unlike those of the Gini index. The bumps that occur on the contours of the Hoover index occur at integer values of the mean as a consequence of the formula for H, the most prominent occurring when the mean is 1. The bumps arise because the Hoover index is not a differentiable (i.e. smooth) function of *m*, which can be seen from an analysis of expression for the Hoover index given in Section 2. These bumps quickly become much less noticeable as the mean increases.

The contour plots of the Gini and Hoover indices exhibit greater differences when considered as functions of *m* and prevalence (Fig 3). First, unlike the Gini index, the contour lines of the Hoover index are parallel to the vertical axis when *m* is less than one. When all infected hosts harbour infrapopulations larger than or equal to the overall mean, the Hoover index is equal to one minus prevalence. For the negative binomial distribution, this implies the Hoover index is equal to one minus prevalence when the mean is less than or equal to one. Second, there is less variability in the widths of the contours for the Hoover index compared to the Gini index. This suggests the dependence of the Hoover index on prevalence is more regular. At a given *m*, a change of 0.1 in the prevalence will have roughly the same effect on the value of the Hoover index, regardless of the initial value of prevalence. In contrast, much of the contour plot of the Gini index is coloured yellow, corresponding to values greater than 0.9. Values of the Gini index less than 0.6 are restricted to small region of the plot, indicating that small changes in prevalence in that region will result in a large change in the Gini index.

The parasite data from Shaw and Dobson were taken from five taxonomic groups. The data, divided into taxa, were superimposed on the plots of *k* vs *m* with contour lines of prevalence and Hoover index. They did not show any obvious grouping.

## 5. Lorenz order and the negative binomial distribution

Both the Hoover and Gini indices are seen in Fig 2 to be decreasing functions of *m* and *k*, as is 1 –prevalence (Fig 1). This behaviour is due to how these indices relate to the Lorenz curve and how the parameters *m* and *k* affect the Lorenz curve of the negative binomial distribution.

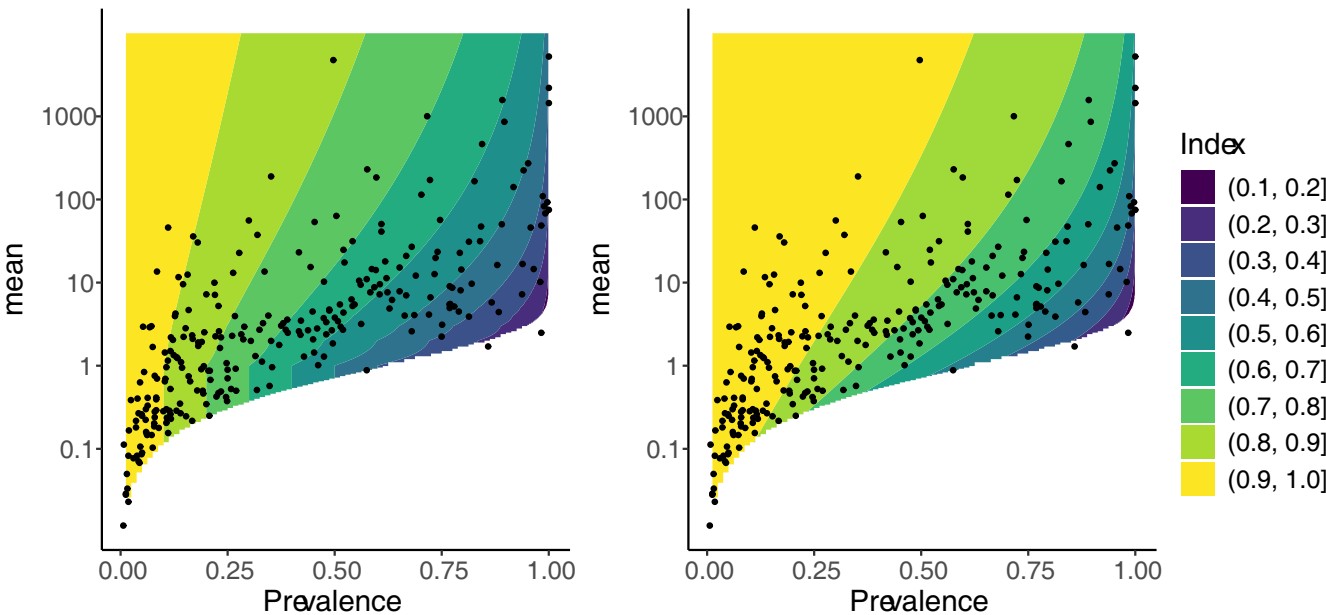

**Fig 3.** Contour plot of Hoover index (left) and Gini index (right) as functions of $m$ and prevalence. The contour lines of the Hoover index are parallel to the vertical axis when $m$ is less than one. The Gini index is less constrained. However, with means above one, the Hoover index is more evenly spread than the Gini index.

The Lorenz curve of a distribution with cumulative distribution function $F$ is given by

$$L(u) = \frac{\int_0^u F^{-1}(y)dy}{m}, u \in [0, 1],$$

where $m$ is the mean of the distribution and $F^{-1}(y) = \sup_x\{x : F(x) \leq y\}$ for $y \in (0, 1)$ [36]. In our context, the Lorenz curve describes the proportion $u$ of the host population that is infected with a proportion $(u)$ of the parasite population. When all hosts have the same parasite burden, the Lorenz curve is given by $(u) = u$ for all $u$ in [0,1]. This is called the egalitarian line. Several indices can be defined in terms of the Lorenz curve. Specifically, the Gini index is twice the area between the Lorenz curve and the egalitarian line, and the Hoover index is the greatest vertical distance between the Lorenz curve and the egalitarian line. Even 1 – prevalence can be viewed as the largest value of $u$ such that $(u) = 0$.

The Lorenz curve induces a partial ordering of distributions. Assume $F_A$ and $F_B$ are two distribution functions with finite means. If the Lorenz curve of $F_A$ is greater than the Lorenz curve of $F_B$ for all $u$, then we say that $F_A$ is smaller than $F_B$ in the Lorenz order and write $F_A \leq_L F_B$. This ordering corresponds to the notion of aggregation put forward by Poulin [18] (see also [9]). From their connections with the Lorenz curve, we see that if $F_A \leq_L F_B$, then the Gini and Hoover indices as well as 1 – prevalence will be smaller for $F_A$ than for $F_B$.

The following result shows how changes to the parameters $m$ and $k$ affect the Lorenz ordering, i.e. the shape of the curve, with a negative binomial distribution.

**Theorem:** Let $NB(m, k)$ denote the negative binomial distribution with parameters $m$ and $k$. If $m_1 < m_2$, then

$$NB(m_2, k) \leq_L NB(m_1, k).$$

If $k_1 < k_2$, then

$$NB(m, k_2) \leq_L NB(m, k_1).$$

The proof is provided in the S2 Supplementary Material in S1 File.

The above result explains why Gini and Hoover indices and 1 −prevalence are all decreasing functions of $m$ and $k$.

Fig 2 also shows that that the contours of both the Gini and Hoover indices become parallel with the axes. This is due to the limiting behaviour of the negative binomial distribution. Depending on how the parameters are allowed to vary, it is known that the negative binomial distribution will converge to either a Poisson distribution or a gamma distribution [37]. Fixing $m$ and letting $k$ increase, the negative binomial distribution converges to a Poisson distribution with mean $m$. This causes the contour lines to become parallel with the horizontal axis as $k$ increases. Similarly, fixing $k$ and letting $m$ increase, an appropriately scaled negative binomial distribution converges to a Gamma distribution with shape and rate parameters both equal to $k$. Since the Gini and Hoover indices are scale invariant [27], these indices will approach their respective values for a Gamma $(k, k)$ distribution as $m$ increases. This causes the contour lines to become parallel with the vertical axis as $m$ increases.

## 6. Discussion

In choosing an index to measure parasite aggregation, indices based on Lorenz curves seem to be favoured, such as the Hoover and Gini as stated above. Compared to the Hoover, our graphs show that the Gini values are closer over a wider range of means, $k$ and prevalence which makes any differences less discernible. For example, for the values of $m$ and $k$ reported in [25], the Gini index exceeds 0.9 in 42% (113/269) of cases compared to 20% (54/269) of cases exceeding 0.9 Hoover index. In addition, the Hoover has a biological interpretation and may be easier to calculate. However, when mean abundances are below one, the Hoover index has restricted values whereas the Gini has no such restriction, suggesting that Gini may be preferred in such a situation. Nevertheless, both indices provide a figure that seems to measure the same phenomenon, a phenomenon that is still undefined.

The graphs provide an easily interpreted demonstration of the effects of the various parameters on the Hoover and Gini indices. Our results demonstrated the deterministic functional relationships between the aggregation indices, and the parameters, mean abundance and prevalence. Values for the indices were calculated directly from the parameters of the negative binomial distribution rather than using simulated data as done by Morrill et al. [23]. This obviated the need to consider the uncertainty of estimates and the effects of different sample sizes.

The relationships were not linear indicating that correlation and principal components analysis may not be the best methods to analyse the relationships. Contour plots are more informative than the single numerical summary given by a correlation. When the relationship between indices is nonlinear, which is the case for those mentioned in Section 2, correlation gives a misleading representation of the strength of the relationship. Though the relationships could be deduced by an analysis of the formulae used to calculate the indices, this is not straightforward. The contour graphs provide an easily accessible way to determine which parameter is having the greatest effect on the index.

In producing the graphs, we assumed that the parasite distributions followed a negative binomial. The contour plots in Fig 3 would remain largely unchanged under small perturbations of the negative binomial distribution, where small is understood in the sense of the Wasserstein distance [38], since the Gini and Hoover indices can be expressed as expectations ([27] Equations 5.5 and 5.35). However, if the distribution is very different to the negative

binomial distribution, then the contour plots may look substantially different. For example, suppose that all hosts have either 0 parasites or n parasites, where n is some fixed positive integer. Then Hoover = Gini = 1 –prevalence. The resulting contour plot of the index as a function of mean and prevalence will just consist of lines parallel to the mean axis, over the region of means and prevalences consistent with this distribution

Listing the advantages and disadvantages of Hoover and Gini indices, Morrill et al [23] describe them as having the disadvantages of being "strongly negatively correlated with prevalence" and "weakly negatively correlated with mean abundance." In contrast, the *k* parameter of the negative binomial distribution and patchiness are described as having the advantages of being "not necessarily correlated with mean abundance" and "only weakly correlated with prevalence." These comments ignore the fact that the negative binomial distribution, and hence any index computed on that distribution, is completely specified by the mean and prevalence. In other words, the dependence of any index on mean and prevalence is perfectly deterministic. In fact, the dependence on any pair of quantities that can be used to parameterise the negative binomial distribution, like *m* and *k* is perfectly deterministic.

Morrill et al [23] argue that of the two indices, the Gini index is to be preferred over the Hoover index on the basis that Hoover index equals one minus prevalence when the mean is less than or equal to one whereas the Gini index has no such restriction. To decide between the Hoover and Gini indices, if one must choose, then the relationship between these indices and *m*, *k* and prevalence need to be considered more closely. Our contour plots (Figs 2 & 3) have shown other differences in the behaviour of the Gini and Hoover indices. Compared to the Gini index, the Hoover index has a greater range over the region of values for *m* and *k* (or prevalence) observed in wild populations and has more regular dependence on prevalence. Given these properties and the Hoover index's clear biological interpretation, we argue that the Hoover index should be preferred over the Gini index, at least when *m* is greater than one.

Our analysis used contour plots to examine how the Gini and Hoover indices are affected by changes in *m*, *k*, and prevalence. This approach could, in principle, be applied to construct contour plots from any three indices, provided two of these can be used to parameterise the negative binomial distribution. For example, one could construct a contour plot of the Gini index as a function of VMR and mean crowding as both *m* and *k* can be expressed in terms of VMR and mean crowding:

$$m = \text{mean crowding} - \text{VMR} + 1$$

and

$$k = \frac{\text{mean crowding}}{\text{VMR} - 1} - 1.$$

The contour plot could then be constructed using the expression for the Gini index in terms of *m* and *k* given in Section 2.

Regardless of how aggregation is measured, aggregation continues to be an important issue in parasitology. Strategies for controlling pests such as leaf minors in coffee plantations need to be modified when aggregation levels are increased, for example due to the use of insecticides [5]. Increased aggregation of helminths in humans may occur after many rounds of chemotherapy so that though the prevalence has decreased to levels no longer considered a public health problem, it is necessary to consider targeting and treating those predisposed to infection or non-compliant to treatment [6, 7, 34].

Clearly measurements of aggregation in some form helps to provide a better picture of parasite population dynamics. Our graphs show that the values from two recommended

techniques using Lorenz curves are disproportionally affected by some parameters more than others. Evaluating the possible influences of means and variances helps to explain differences in aggregation and may lead to improved management.

## Supporting information

**S1 File.**
(PDF)

## Acknowledgments

We thank anonymous reviewers for their useful comments

## Author Contributions

**Conceptualization:** R. McVinish, R. J. G. Lester.

**Formal analysis:** R. McVinish.

**Methodology:** R. McVinish, R. J. G. Lester.

**Visualization:** R. J. G. Lester.

**Writing – original draft:** R. McVinish, R. J. G. Lester.

**Writing – review & editing:** R. McVinish, R. J. G. Lester.

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
