## [Decision Letter · Decision Letter 0]

23 Sep 2024

PONE-D-24-29289A graphical exploration of the relationship between parasite aggregation indices.PLOS ONE

Dear Dr. Lester,

Thank you for submitting your manuscript to PLOS ONE. After careful consideration, we feel that it has merit but does not fully meet PLOS ONE’s publication criteria as it currently stands. Therefore, we invite you to submit a revised version of the manuscript that addresses the points raised during the review process.

We look forward to receiving your revised manuscript.

Kind regards,

Clement Ameh Yaro, Ph.D

Academic Editor

PLOS ONE

Journal Requirements:

Reviewers' comments:

Reviewer's Responses to Questions

**Comments to the Author**

1. Is the manuscript technically sound, and do the data support the conclusions?

Reviewer #1: Yes

Reviewer #2: Yes

Reviewer #3: Yes

Reviewer #4: Yes

Reviewer #5: Yes

Reviewer #6: Yes

2. Has the statistical analysis been performed appropriately and rigorously? 

Reviewer #1: Yes

Reviewer #2: Yes

Reviewer #3: Yes

Reviewer #4: Yes

Reviewer #5: I Don't Know

Reviewer #6: Yes

3. Have the authors made all data underlying the findings in their manuscript fully available?

Reviewer #1: Yes

Reviewer #2: Yes

Reviewer #3: Yes

Reviewer #4: Yes

Reviewer #5: Yes

Reviewer #6: Yes

4. Is the manuscript presented in an intelligible fashion and written in standard English?

Reviewer #1: Yes

Reviewer #2: Yes

Reviewer #3: Yes

Reviewer #4: Yes

Reviewer #5: Yes

Reviewer #6: Yes

5. Review Comments to the Author

Reviewer #1: This manuscript attempts to clarify the relationship between most common used indices of aggregation in parasitic systems studies with mean abundance and prevalence, by using contour plots. Contour plots analysis have been rarely used in parasitology, at my best knowledge. However, this manuscript provides a simple and insightfull way to explore and comprehend relationships between aggregation and the widely calculated parameters of host-parasitic studies. Aggregation of parasites in population of host is a well known almost universal characteristic, but seldom analyzed. However, analyses herewith presented in this manuscript would help in choosing an index to use to measure aggregation; Hoover and Gini indices are favored by authors. Further authors discuss what parameter is having the greatest effect on the index which is most useful to understand biological relationships. A highlight point is also the calculations and analyses are made under R ambient, this facilitated a wide usage of the author’s recommendations. Clearly, this Ms deserves publication in Plos One.

Reviewer #2: This is an interesting research, so I recommend to the editor for the acceptance of the paper because it add largely to the body of scientific knowledge and potentially will be useful for future study.

Reviewer #3: Overall Comments:

This manuscript presents an interesting graphical exploration of relationships between different parasite aggregation indices using contour plots. The approach is novel and provides valuable insights into how these indices relate to each other and to parameters of the negative binomial distribution. However, there are several areas that require revision before the manuscript is suitable for publication, including improving clarity of explanations, addressing some potential scientific issues, and enhancing the figures.

Major Comments:

1. Introduction:

- Lines 14-16: The introduction of aggregation indices is abrupt. Provide a brief explanation of why these indices are important in parasitology.

- Lines 22-24: Expand on why contour plots provide a "more accurate representation" than correlations. What specific insights do they offer?

-You need to suopprt your introduction with some references regarding the dynamics of parasites:

-https://www.nature.com/articles/s41598-024-70528-x

-https://pubmed.ncbi.nlm.nih.gov/39039589/

2. Methods:

- Lines 61-63: The explanation of how the contour plots were constructed is unclear. Provide more detail on the mathematical process.

- Lines 70-72: Clarify how regions inconsistent with a negative binomial distribution were determined.

- Line 76: Provide justification for using log scaling for m and k.

3. Results:

- Section 3: This section mixes results and interpretation. Consider separating pure results from their interpretation.

- Lines 90-92: The statement about the relationship between m and k after controlling for prevalence needs further explanation or evidence.

- Lines 134-136: The claim about the stability of indices over a wide range of m and k needs quantification or clearer illustration.

4. Discussion:

- Lines 236-238: The conclusion about preference for the Hoover index over the Gini index needs more robust justification.

- Lines 247-250: The possibility of constructing contour plots from any three indices is interesting but needs more explanation on its potential applications and limitations.

5. Figures:

- Figure 1: The color scheme makes it difficult to distinguish between different prevalence levels, especially in the mid-range. Consider using a different color palette.

- Figures 2 and 3: Add labels to the contour lines to make the values easier to read.

- All figures: Increase font size for axis labels and legends to improve readability.

Minor Comments:

1. Line 19: "Poulin's D" is mentioned in keywords but not explained in the text. Either remove or explain.

2. Line 46: "two other indices are derived" - specify that these are the Gini and Hoover indices.

3. Line 82: "reported prevalences" should be "reported prevalence values".

4. Line 155: "bumps that occur on the contours" - explain what causes these bumps.

5. Line 199: "decreases in the Lorenz order" - briefly explain what this means for readers unfamiliar with the concept.

6. Line 244: "unravel other complex relationships" - provide an example to illustrate this point.

Questions for Authors:

1. How sensitive are your results to the assumption of a negative binomial distribution? Would the relationships between indices be substantially different for other distributions?

2. Can you provide guidance on when researchers should prefer contour plots over correlation analysis when examining relationships between aggregation indices?

3. How might the insights from your contour plots be applied in practical parasitological research or wildlife management?

4. Are there situations where the differences between the Gini and Hoover indices become particularly important? Can you elaborate on when one might be preferable over the other?

5. How do your findings relate to or extend previous work on the relationships between aggregation indices in parasitology?

In conclusion, while this manuscript presents a novel and potentially valuable approach to understanding parasite aggregation indices, major revisions are required to improve clarity, address potential scientific issues, and enhance the presentation of results. The authors should carefully address all points raised above in their revision.

Reviewer #4: In this manuscript, the authors have provided a thorough analysis into parasite population dynamics, exploring the concept of parasite aggregation which is poorly defined. By providing a more accurate representation, the manuscript attempts to assist scientists in understanding the concept better. Overall, the authors have done a good job in presenting the data in a detailed yet easy way, for the reader to grasp.

Reviewer #5: Comments

The manuscript entitled " A graphical exploration of the relationship between parasite aggregation indices" found to be very interesting study that well describe, analysed and written.

Please check and revise the manuscript.

Reviewer #6: The introduction, methods and materials, results and conclusions have been well written.

The graphical representations have also been well presented. Overall, the paper will have greater readership.

6. PLOS authors have the option to publish the peer review history of their article (what does this mean?). If published, this will include your full peer review and any attached files.

Reviewer #1: No

Reviewer #2: No

Reviewer #3: No

Reviewer #4: No

Reviewer #5: **Yes: **Magda Elsayed Abd-Elgawad

Reviewer #6: No

---

## [Author Response · Author response to Decision Letter 0]

19 Nov 2024

Dear Clement Ameh Yaro,

 I think we have the manuscript in the correct form for submission and have addressed the reviewer comments. Thank you for your consideration.

---

## [Decision Letter · Decision Letter 1]

2 Dec 2024

A graphical exploration of the relationship between parasite aggregation indices.

PONE-D-24-29289R1

Dear Dr. Lester,

We’re pleased to inform you that your manuscript has been judged scientifically suitable for publication and will be formally accepted for publication once it meets all outstanding technical requirements.

Kind regards,

Clement Ameh Yaro, Ph.D

Academic Editor

PLOS ONE

Additional Editor Comments (optional):

Reviewers' comments:

Reviewer's Responses to Questions

**Comments to the Author**

1. If the authors have adequately addressed your comments raised in a previous round of review and you feel that this manuscript is now acceptable for publication, you may indicate that here to bypass the “Comments to the Author” section, enter your conflict of interest statement in the “Confidential to Editor” section, and submit your "Accept" recommendation.

Reviewer #3: All comments have been addressed

2. Is the manuscript technically sound, and do the data support the conclusions?

Reviewer #3: Yes

3. Has the statistical analysis been performed appropriately and rigorously? 

Reviewer #3: Yes

4. Have the authors made all data underlying the findings in their manuscript fully available?

Reviewer #3: Yes

5. Is the manuscript presented in an intelligible fashion and written in standard English?

Reviewer #3: Yes

6. Review Comments to the Author

Reviewer #3: I accept it for publication after authors addressing all my comments, the revised version is now suitable for publication

7. PLOS authors have the option to publish the peer review history of their article (what does this mean?). If published, this will include your full peer review and any attached files.

Reviewer #3: No

---

## [Editor Report · Acceptance letter]

17 Dec 2024

PONE-D-24-29289R1 

PLOS ONE

Dear Dr. Lester, 

I'm pleased to inform you that your manuscript has been deemed suitable for publication in PLOS ONE. Congratulations! Your manuscript is now being handed over to our production team.

Kind regards, 

on behalf of

Dr. Clement Ameh Yaro 

Academic Editor

PLOS ONE